# Role of individual and population heterogeneity in shaping dynamics of multi-pathogen shedding in an island endemic bat

Samantha Aguillon[¤a]*, Magali Turpin, Gildas Le Minter, Camille Lebarbenchon, Axel O. G. Hoarau, Céline Toty, Avril Duchet, Léa Joffrin[¤b], Riana V. Ramanantsalama[¤c], Pablo Tortosa, Patrick Mavingui, Muriel Dietrich*

UMR PIMIT (Processus Infectieux en Milieu Insulaire Tropical), Université de la Réunion/ INSERM1187/ CNRS9192/ IRD249, Sainte-Clotilde, Reunion Island

¤a Current address: Chrono-environnement, UMR 6249/CNRS, Université Marie et Louis Pasteur, Besançon, France
¤b Current address: Evolutionary Ecology Group, Department of Biology, University of Antwerp, Antwerp, Belgium
¤c Current address: Behavioral Ecology & Sociobiology Unit, German Primate Center, Göttingen, Germany
* samantha.aguillon@univ-reunion.fr (SA); muriel.dietrich@ird.fr (MD)

## Abstract

Understanding processes driving pathogen transmission in bats is critical to prevent spillovers and emergence events. Although substantial research has addressed this topic, few studies have directly examined shedding dynamics (as opposed to serological studies) and co-infection patterns using fine-scale spatio-temporal data-sets. Here, based on the monitoring of 5,714 Reunion free-tailed bats (*Mormopterus francoismoutoui*) in 17 roosts over 24 months, we studied the co-shedding dynamics of paramyxoviruses (PMV) and *Leptospira* bacteria (LEPTO) in urine, and herpes-viruses (HSV) in saliva. We evidenced all year long shedding with high prevalence of all three infectious agents (37% - 87%), as well as an exceptionally high level of co-shedding (59%), with both positive and negative interactions between infectious agents. Shedding patterns displayed temporal synchrony among roosts, with a peak during summer months, but were not influenced by roost size. Repeated shedding in recaptured bats supports within-host persistence, though underlying mechanisms remain to be identified. Our results also showed rapid HSV infection of juveniles (< 6 months), and suggest longer protection of juveniles by maternal antibodies for PMV and LEPTO. Reproductively-active individuals (both during the pregnancy and mating) were associated with increased PMV and LEPTO shedding, which could result from tradeoffs between reproduction and infection in both sexes, and/or an age-related bias with the progressive infection of older juveniles during reproductive periods. This study highlights the significance of persistent shedding of multiple pathogens, including bacteria, and their intricate interactions within bat populations. Understanding how human-driven ecological changes may disrupt within-host

**Data availability statement:** Metadata are archived on Zenodo (https://doi.org/10.5281/zenodo.14842155).

**Funding:** This research was supported by the French National Research Agency (ANR-17-CE35-0008-01 to MD) and by the European Regional Development Funds (ERDF PO INTERREG V ECOSPIR, number RE6875 to PT). SA and AOGH were supported by a "Contrat Doctoral de l'Université de La Réunion" from the Minister of Higher Education, Research and Innovation and LJ by an "Allocation Régionale de Recherche de la Région Réunion". The funders had no role in study design, data collection and analysis, decision to publish, or preparation of the manuscript.

**Competing interests:** The authors have declared that no competing interests exist.

processes and influence pathogen shedding in bats will help assessing the risk of pathogen spillover from bats to other species, including humans.

## Author summary

Understanding risks of bat-borne pathogen spillover is challenging because of the difficulty in studying shedding dynamics in wild bat populations. Here, we used an original island-endemic bat species to build up a fine-scale spatio-temporal shedding analysis across the entire island. By studying two viruses (paramyxoviruses and herpesviruses) and a bacterium (*Leptospira*), we conducted analyses at both population- (roost) and individual- (through recaptured bats) levels. Although infection occurs year-round, shedding patterns display a temporal synchrony with seasonal peaks in prevalence. These patterns are driven by the age of bats and associated with the reproductive periods in both females (pregnancy) and males (mating). Results also suggest that persistence, clearance cycles, as well as interactions between infectious agents (antagonistic or facilitative), are important within-host processes that contribute to the transmission of infections in bat populations. More research is essential to understand how human activities may influence these co-shedding patterns and the risk of cross-species transmission.

## Introduction

Several human outbreaks, such as those caused by the SARS-CoV-1 and Nipah viruses, have highlighted the importance of understanding processes driving the transmission of infectious agents within bat populations [1]. Indeed, infection dynamics in bat reservoir hosts have been linked to zoonotic spillovers [2,3]. Thus, identifying the ecological drivers of pathogen transmission within bat populations, at both within- and between-host levels, is a key element for predicting spillover events [4,5]. Unfortunately, ecological studies investigating infection dynamics in bats often lack robust spatio-temporal datasets [6] (but see, e.g., [7–10]). This gap mainly results from logistical constraints due to the elusive and roaming nature of bats that make these animals difficult to study over long periods, and limits the breadth of data acquisition. Moreover, these gregarious mammals often display metapopulation dynamics [11]. Such spatio-temporal heterogeneity is characterized by periodically interacting spatial discrete patches, with various sex and age-structure, roost size and function [12,13], which adds complexity to disentangle individual and populational drivers of infection dynamics in bats [8,14,15].

 With molecular tools becoming increasingly accessible for the study of infectious agents, it is now recognized that co-infections are common in bats [16–21]. However, most of the previous longitudinal studies on bats have focused on a single infectious agent, or different viral species within the same family (e.g., [2,22–24]). Thus, our

knowledge on spatio-temporal co-infection dynamics with different groups of infectious agents (DNA/RNA viruses, bacteria, fungi) remains limited [10,25–27]. Theory predicts that co-infections can have major consequences on within-host infection dynamics because of potential interactions that might be antagonistic or mutualistic [19,28,29]. For example, the infection of field voles with the bacterium *Anaplasma phagocytophilum* increased by five times the susceptibility to cowpox virus [30]. Examining the spatio-temporal patterns of simultaneous shedding of infectious agents within bat roosts (co-circulation) and individual hosts (co-infection) is likely to provide much-needed further insights into underlying processes driving infection dynamics [2].

Islands are widely considered as relevant model systems in ecology and evolutionary biology studies [31,32], including those focusing on emerging diseases (e.g., [33,34]). These ecosystems typically support a limited number of species and exhibit high levels of endemism [35]. Due to their ability to fly over water, bats are often the only native mammals in remote oceanic islands. Among the 1,456 recognized bat species, 60% inhabit islands and 25% are even island-endemic [36]. The clearly defined geographical boundaries and often reduced surface of islands enable sampling endemic bat populations at a fine scale, by focusing sampling efforts on a specific limited area which covers a comprehensive range of sites, and this is crucial to make reliable inferences about spatial and temporal infection dynamics [37]. Moreover, the reduced number of bat species on islands may also simplify ecological interactions, by minimizing interspecies contacts, and by sampling monospecific bat roosts, which can contribute to a more in-depth understanding of infection dynamics at the population level [38]. Due to the reduced species richness typical of small territories, insular ecosystems may also lack predators or competitors and offer empty ecological niches, which can favor the spread of island endemic bats and the expansion of their population size, although this would ultimately be constrained by the availability of resources. This is especially true for bat families with highly gregarious behavior such as free-tailed bats (Molossidae), for which roosts can host hundreds of thousands of individuals [39,40]. High bat densities favor the sampling of large numbers of individuals, a prerequisite for epidemiological studies. Island endemic bats, with such an abundant, simplified, and well-defined system, represent a rare opportunity to collect extensive spatio-temporal data to capture a comprehensive picture of subtle mechanisms underlying infection dynamics in bats [38].

The Reunion free-tailed bat (*Mormopterus francoismoutoui*; [41]) is a small tropical insectivorous bat endemic to Reunion Island. This volcanic territory is located in the southwestern Indian Ocean and emerged about 3 million years ago [42]. Although small in size (2,512 km²), Reunion Island is shaped by a mountainous landscape and has undergone significant landscape changes following 350 years of human colonization associated with intensive urbanization and agriculture [43]. Reunion free-tailed bats are broadly distributed on the island and monospecific roosts occur in a diversity of natural habitats, such as caves and cliffs. These mollosid bats have also adapted to urbanization and use numerous anthropogenic roosts such as buildings and bridges, hosting from a few hundred to more than 100,000 individuals [12,41]. A recent longitudinal monitoring of several roosts revealed highly dynamic roosting behaviors, with roosts of various sizes, sex-ratio, and functions (female-biased maternities, mixed colonies or male-biased roosts) [12]. Specifically, large female aggregations (up to 50,000 pregnant females) are observed synchronously every year in a few roosts, before parturition in austral summer (December), which coincides with a female-biased sex-ratio and a spatial sexual segregation during this period of the year [12,44]. Moreover, the end of austral summer (May) is characterized by a decrease in roost size and a change in sex-ratio in the studied roosts (from female- to male-biased), suggesting important seasonal sex-specific movements within the island, which is corroborated by the absence of spatial genetic structure in this metapopulation [45].

Reunion free-tailed bats are natural hosts for several infectious agents [10,22,46], including paramyxoviruses and *Leptospira* bacteria [44]. Previous non-invasive sampling of bat faeces and urine underneath roosts highlights that shedding pulses of paramyxoviruses and *Leptospira* bacteria occurred during the austral summer, followed by a sustained transmission during winter [10,44,46]. However, these studies relied on a reduced number of samples that did not include information on individual bats (as no captures were performed), precluding direct insights into how individual factors (e.g., sex, age, reproductive status) influence these shedding dynamics. Moreover, studies were conducted on only two maternity

roosts [10,44] and the role of population-level factors (e.g., roost size, sex-ratio) thus remains poorly understood. Here, we used data from a fine-scale spatio-temporal monitoring and sampling of *M. francoismoutoui*, based on the study of 17 roosts all over Reunion Island during two consecutive years. We measured the simultaneous shedding (nucleic acid detection) of three infectious agents in each individual bat, focusing on paramyxoviruses (viral RNA) and *Leptospira* (bacteria DNA) both shed in urine, and herpesviruses (viral DNA) in the saliva. Specifically, we first aimed to test for spatio-temporal trends in shedding prevalence and synchrony across roost sites. Then, we evaluated whether individual (sex, age, reproductive status) and roost-level (roost size and level of sexual spatial segregation) variables explained prevalence of single infectious agents, co-shedding patterns, and the probability of changing shedding status (within-host dynamics).

## Results

### Spatio-temporal dynamics of shedding prevalence

Between January 2019 and December 2020 (24 months), we captured 5,714 Reunion free-tailed bats in 17 roosts all over Reunion Island (Fig 1), including 405 recaptures (up to four recaptures were recorded for the same individual). Shedding data were obtained from 5,518 urine samples for paramyxoviruses (PMV) and *Leptospira* bacteria (LEPTO), and from 3,981 saliva samples for herpesviruses (HSV), including 3,784 individuals for which all three infectious agents were

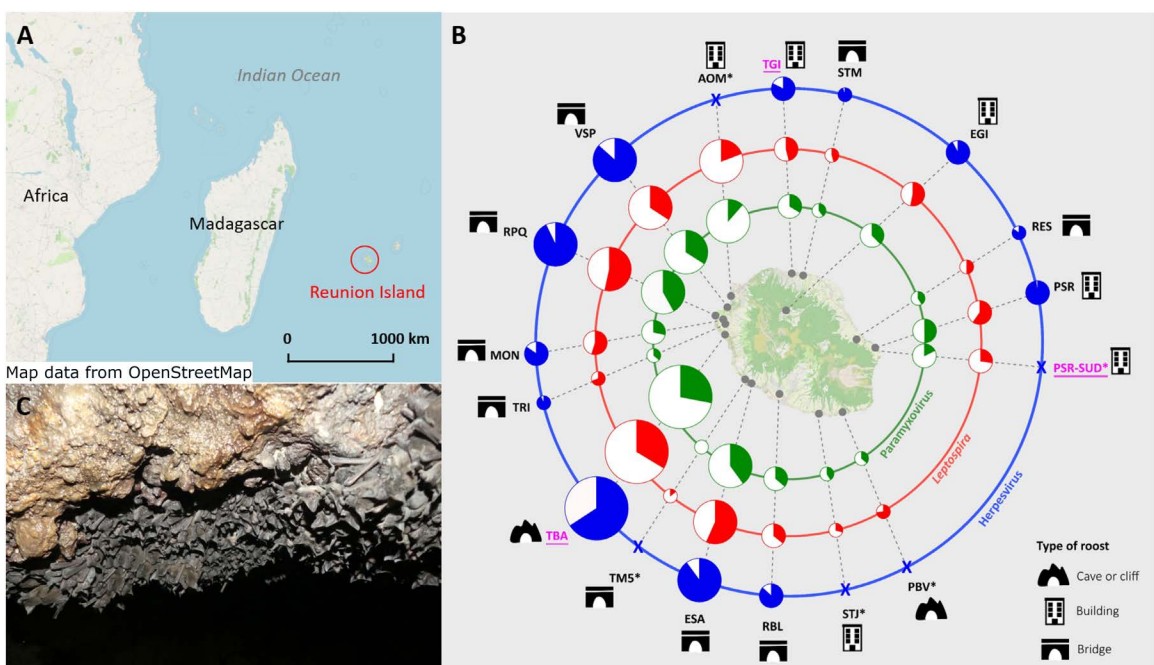

**Fig 1. Widespread triple shedding in *Mormopterus francoismoutoui*.** (A) Location of Reunion Island and (B) the studied roosts. (C) A picture of juvenile bats aggregated in the maternity roost (TBA). In (B) studied roosts are represented by grey circles on Reunion Island map and prevalence (color section of pie charts) is represented for paramyxoviruses (in green), *Leptospira* bacteria (in red) and herpesviruses (in blue). A cross denotes the absence of prevalence data (no tested samples). Pie diameter corresponds to four categories of roost size: small < 500 individuals, medium: 500-2000 individuals, high: 2000-6000 individuals, and exceptional > 100,000 individuals. Accurate roost size estimation was not being performed for EGI and PSR roosts, but they are represented as medium roosts because several hundred/thousands are expected. Maternity roosts are underlined in pink. Roosts that were monitored only once are depicted by an asterisk. Maps were created using Qgis [112] with a background of OpenStreetMap (https://www.openstreetmap.org) under the Open Data Commons Open Database License (https://www.openstreetmap.org/copyright). This figure was inspired by Streicker et al. 2012 [53].

tested. Over the entire study period, we measured a high global prevalence with 36.63% ± 1.27% of bats shedding PMV, 46.48% ± 1.32% shedding LEPTO and 87.14% ± 1.04% shedding HSV (considering recaptured bats as independent points in time) (Table 1).

Shedding of the three infectious agents was detected in all tested roosts, except PMV in the TM5 roost (temporary roost exclusively composed of juveniles; see [12]). Pairwise comparison of prevalence among roosts at different sampling periods (calculated as Pearson's correlations) revealed both positive and negative correlation coefficients, and thus a moderate level of spatial synchrony, which was slightly higher for HSV (regional synchrony = 0.269) compared to PMV (regional synchrony = 0.151) and LEPTO (regional synchrony = 0.202). Moreover, spline correlograms showed that synchrony was not influenced by the geographical distance between roosts (Fig 2A). A dynamic factor analysis (DFA) was then used to model common trends in prevalence across roosts. Despite variation of prevalence patterns among roosts (Fig 2B), DFA models converged and showed a common trend that peaked during the summer months (from October to December) for the three infectious agents, and then reached its lowest around March to June (Fig 2C). Roost loadings varied substantially in both magnitude and direction. In particular, the roosts MON, RPQ and TGI showed the lowest loadings (sometimes negative) which implies some degree of asynchrony for these roosts. Finally, the use of generalized additive models (GAM) (S1 and S3 Tables), to characterize variation in prevalence, confirmed significant intra-annual variations for PMV and HSV prevalence (model M1: $\chi^2_{5.7} = 15.05$, $P = 0.03$; model M3: $\chi^2_{6.7} = 19.86$, $P = 0.01$; S1 Table).

**Table 1. Details on the sampled roosts of *M. francoismoutoui* and the number of tested and positive samples. Detections for each pathogen are reported at the family (paramyxoviruses, herpesviruses) or genus (Leptospira) level, and may represent combined prevalences across multiple species.**

| Roost | Habitat | No. of sampling periods | Roost size estimation (min and max when available) | No. of urine tested | No. of PMV positive (%) | No. of LEPTO positive (%) | No. of saliva tested | No. of HSV positive (%) |
|---|---|---|---|---|---|---|---|---|
| AOM | Building | 1 | 2560 | 52 | 6 (11.5) | 10 (19.2) | 0 | no data |
| EGI | Building | 3 | no data | 149 | 56 (37.6) | 79 (53.0) | 89 | 82 (92.1) |
| ESA | Bridge | 15 | 570 - 3200 | 634 (PMV) 635 (LEPTO) | 254 (40.0) | 360 (56.7) | 452 | 406 (89.9) |
| MON | Bridge | 15 | 210 - 1150 | 488 | 144 (28.4) | 268 (54.9) | 419 | 355 (84.5) |
| PBV | Cliff | 1 | 300 | 11 | 4 (36.4) | 8 (72.7) | 0 | no data |
| PSR | Building | 15 | no data | 512 (PMV) 510 (LEPTO) | 255 (49.8) | 305 (59.8) | 396 | 387 (97.8) |
| PSR-SUD | Building | 1 | no data | 51 | 9 (17.6) | 14 (27.4) | 0 | no data |
| RBL | Bridge | 15 | 350 - 1300 | 658 | 238 (36.2) | 235 (35.7) | 425 | 369 (86.8) |
| RES | Bridge | 8 | 35 - 410 | 223 | 91 (41.0) | 111 (49.8) | 212 | 179 (84.4) |
| RPQ | Bridge | 15 | 150 - 3000 | 535 (PMV) 536 (LEPTO) | 229 (41.9) | 287 (53.4) | 416 | 386 (92.8) |
| STJ | Building | 1 | 200 | 34 | 14 (41.2) | 10 (29.4) | 0 | no data |
| STM | Bridge | 12 | 27 - 350 | 364 | 145 (39.9) | 166 (45.6) | 281 | 267 (95.0) |
| TBA* | Cave | 11 | 0 - 100,000 | 500 | 140 (28.0) | 168 (33.6) | 330 | 217 (65.8) |
| TGI* | Building | 15 | 75 - 1270 | 496 | 167 (33.7) | 229 (46.1) | 432 | 356 (82.4) |
| TM5 | Bridge | 1 | 200 | 28 | 0 | 4 (14.3) | 0 | no data |
| TRI | Bridge | 4 | 153 - 251 | 125 | 45 (36.0) | 86 (68.8) | 112 | 104 (92.9) |
| VSP | Bridge | 15 | 15 - 6000 | 658 | 224 (34.0) | 225 (34.2) | 417 | 361 (86.6) |
| Total | / | / | / | 5,518 | 2,021 (36.6) | 2,565 (46.5) | 3,981 | 3,469 (87.1) |

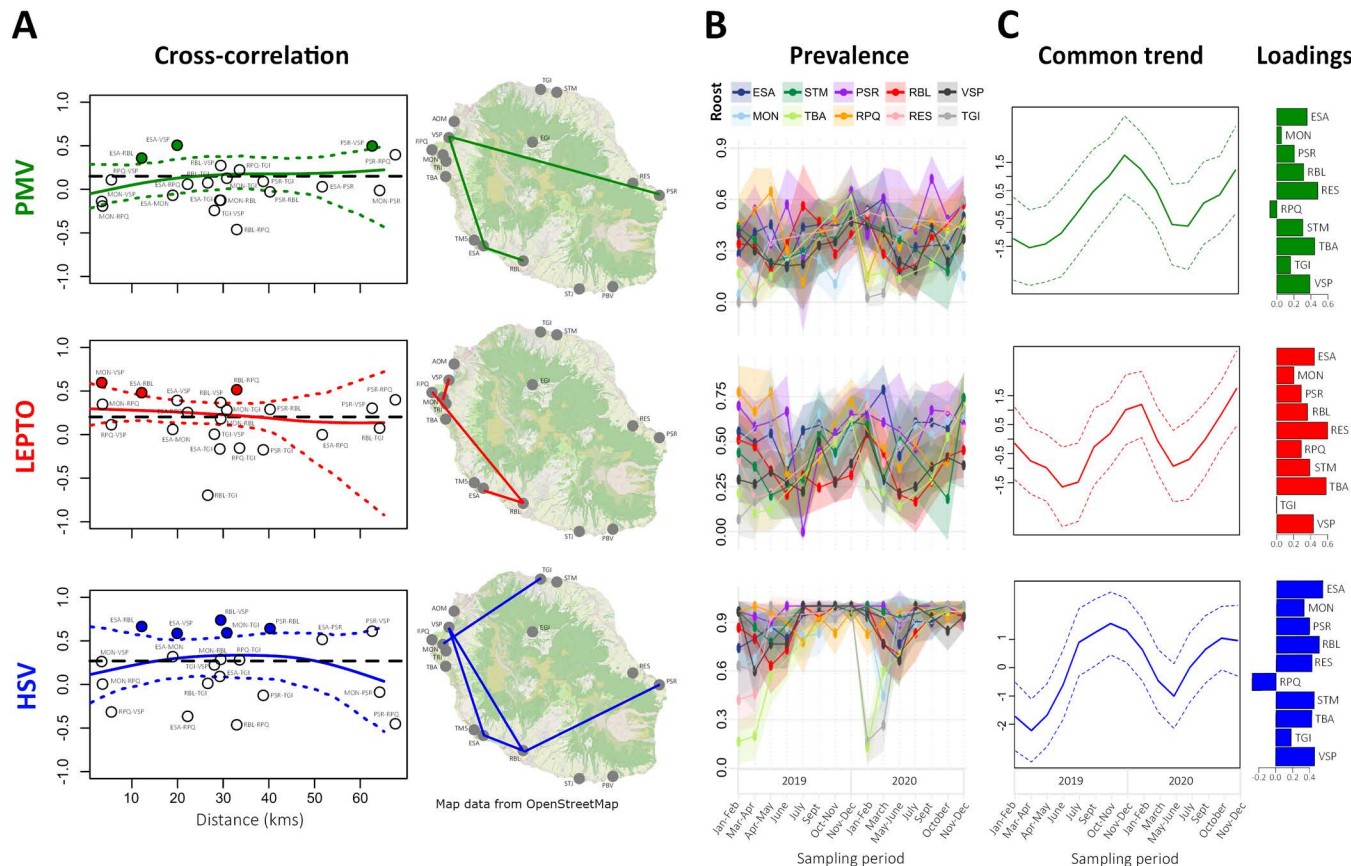

**Fig 2. Synchrony of prevalence across roosts for paramyxoviruses (PMV), *Leptospira* bacteria (LEPTO) and herpesviruses (HSV) in *M. francoismoutoui*.** (A) Pairwise Pearson's correlation coefficients are depicted in relation to geographical distance between roosts. Coefficients were not calculated for TBA, RES, and STM roosts because of missing data during winter (these roosts were empty at this time). The spline correlogram (thick colored line) and 95% confidence interval (dotted colored lines) are shown. The mean estimated synchrony is represented by the black dotted line. Positive correlation coefficients that exceed the spline's confidence interval are highlighted with colored circles and are represented on the map by connecting lines between roosts. (B) Roost-specific raw prevalence across time (thick line) and 95% confidence interval (colored band). (C) Estimated common prevalence trend (thick line) across time and confidence interval (dotted lines). Loadings describing the relationships between roosts and the common trend are shown on the right. Three-letter code refers to roost name as in Table 1. Maps were created using Qgis [112] with a background of OpenStreetMap (https://www.openstreetmap.org/) under the Open Data Commons Open Database License (https://www.openstreet-map.org/copyright).

## Roost-level drivers of shedding

Roost size varied from 15 to 100,000 individuals but was not correlated with the prevalence of PMV and LEPTO (model M1: $z = 0.37$, $P = 0.71$; model M2: $z = 0.80$, $P = 0.43$; S1 Table), while HSV prevalence slightly decreased in larger roosts (model M3: $z = -2.83$, $P = 0.005$; S1 Table). However, this negative correlation was driven by the presence of juveniles when roost sizes were the highest, these juveniles being low infected (see below). This negative correlation disappeared when juveniles were removed from the HSV model (model M3bis: $z = -0.83$, $P = 0.41$; S1 Table). Because adult males and females aggregate and segregate differently among roosts [12], we also examined the role of sexual segregation levels of adults, but did not find any correlation with prevalence (model M1: $\chi^2_{4.3} = 1.55$, $P = 0.90$; model M2: $\chi^2_{2.9} = 1.76$, $P = 0.60$; model M3: $\chi^2_{4.9} = 7.38$, $P = 0.40$; S1 Table).

## Individual variation of shedding

Models revealed that adults were systematically more frequently shedding viruses and bacteria than juveniles (model M1: $z = -5.83$, $P = 5.62^{-09}$; model M2: $z = -6.17$, $P = 6.73^{-10}$; model M3: $z = -11.88$, $P < 2^{-16}$; S1 Table). Moreover, for *Leptospira* bacteria that was detected using a real-time PCR, we analyzed the cycle threshold (Ct) values which serve as a proxy for bacterial load. The model indicated that, when infected, juveniles shed a significantly lower *Leptospira* load compared to adults (model M4: $z = 11.78$, $P < 2^{-16}$; S1 Table).

Mean shedding prevalence in adults was high: $43.00 \pm 1.42\%$ for PMV, $52.85 \pm 1.43\%$ for LEPTO and $96.20 \pm 0.65\%$ for HSV (Fig 3A–3C). In contrast, at the time when juveniles started to fly (January-February; [12]), shedding prevalence in juveniles was either null (PMV) or low (LEPTO: $12.03 \pm 4.11\%$ and HSV: $15.03 \pm 5.66\%$). Then, over the following six months, HSV prevalence increased drastically in juveniles to reach $60.67 \pm 7.82\%$, while it remained close to 0% for PMV, or only slightly increased for LEPTO (Fig 3A–3C). After six months, we could not monitor properly infection in juveniles because most of bats were classified as adults based on the epiphysis fusion, but we noted that for females with non-developing nipples (M0 females; that probably include juveniles), PMV and LEPTO infection remained low for at least 6 more months (S1 Fig). Shedding prevalence also varied according to sex, but only in adult bats, and differently according to the targeted infectious agent (model M1: $z = -5.03$, $P = 4.81^{-07}$; model M2: $z = -0.77$, $P = 0.44$; model M3: $z = 3.84$, $P = 1.25^{-04}$; S1 Table) (Fig 3D–3F). Indeed, adult males were more frequently shedding HSV than females (Tukey's post hoc test: $t = -3.84$, $P = 7.0^{-04}$), although the opposite pattern was observed for PMV (Tukey's post hoc test, $t = 5.03$, $P < 1.0^{-04}$).

To test the influence of reproduction on the shedding status, we focused our analysis on adults during two reproductive periods: pregnancy (November-December) and mating (March-April). During the pregnancy period, the detection of the two urine-excreted agents, PMV and LEPTO, was higher in pregnant than in non-pregnant females (model M5: $z = 5.76$, $P = 8.31^{-09}$; model M6: $z = 5.20$, $P = 2.02^{-07}$; S2 Table) (Fig 3D, 3E). However, when we excluded non-pregnant females with no visible nipples (M0) from the models—potentially representing misclassified old juveniles—the effect of pregnancy was no longer detected (model M5bis: $z = 0.72$, $P = 0.47$; model M6bis: $z = 1.35$, $P = 0.18$; S2 Table) (Fig 3D, 3E). Moreover, during November-December, non-pregnant females without visible nipples were indeed less frequently infected compared to non-pregnant females with visible nipples, regardless of pregnancy status (S1 Fig). For males, during the mating period, reproductively active individuals were more frequently found shedding LEPTO than non-reproductive males (model M10: $z = 2.70$, $P = 0.007$; S2 Table) (Fig 3E). Finally, we did not find any influence of reproduction of the *Leptospira* load (Ct) for both females and males (model M7: $z = -1.69$, $P = 0.09$; model M11: $z = 1.62$, $P = 0.11$; S2 Table).

## Co-shedding dynamics

Among the 3,784 bats with paired samples (both urine and saliva collected), 36.89% (n = 1,396) were shedding two infectious agents and 22.46% (n = 850) were shedding three (S2 Fig). Only 399 bats (10.54%) did not excrete any of the three screened infectious agents, with 79.2% of them being noteworthily juveniles. Shedding one infectious agent increased the probability of shedding a second infectious agent (for all combinations in models M1, M2, M3, all $P < 0.001$; S1 Table) (Fig 4). The relationship between viral infections was stronger than that between viruses and bacteria. For example, based on odd ratios estimated from models M1-M3, bats shedding HSV were 14 times more likely to shed PMV, whereas shedding HSV increased the likelihood of LEPTO shedding only by threefold. Additionally, individuals already shedding two infectious agents were more likely to shed a third (Fig 4). However, in cases of triple infections, a negative interaction was observed, indicating that the combined effect of two agents on the prevalence of the third was weaker than expected if the two infections were independent (model M1: $z = -3.10$, $P = 0.002$; model M2: $z = -2.46$, $P = 0.01$; model M3: $z = -2.51$, $P = 0.01$; S1 Table).

Models showed no significant intra-annual variations for dual and triple shedding, except for PMV-HSV (model M15: $\chi_{5.7} = 17.46$, $P = 0.01$; S3 Table), and no effect either of sexual segregation or roost size. By contrast, we detected a strong influence of age, with dual and triple shedding being more frequent in adults (model M13: $z = -4.73$, $P = 2.30^{-06}$; model

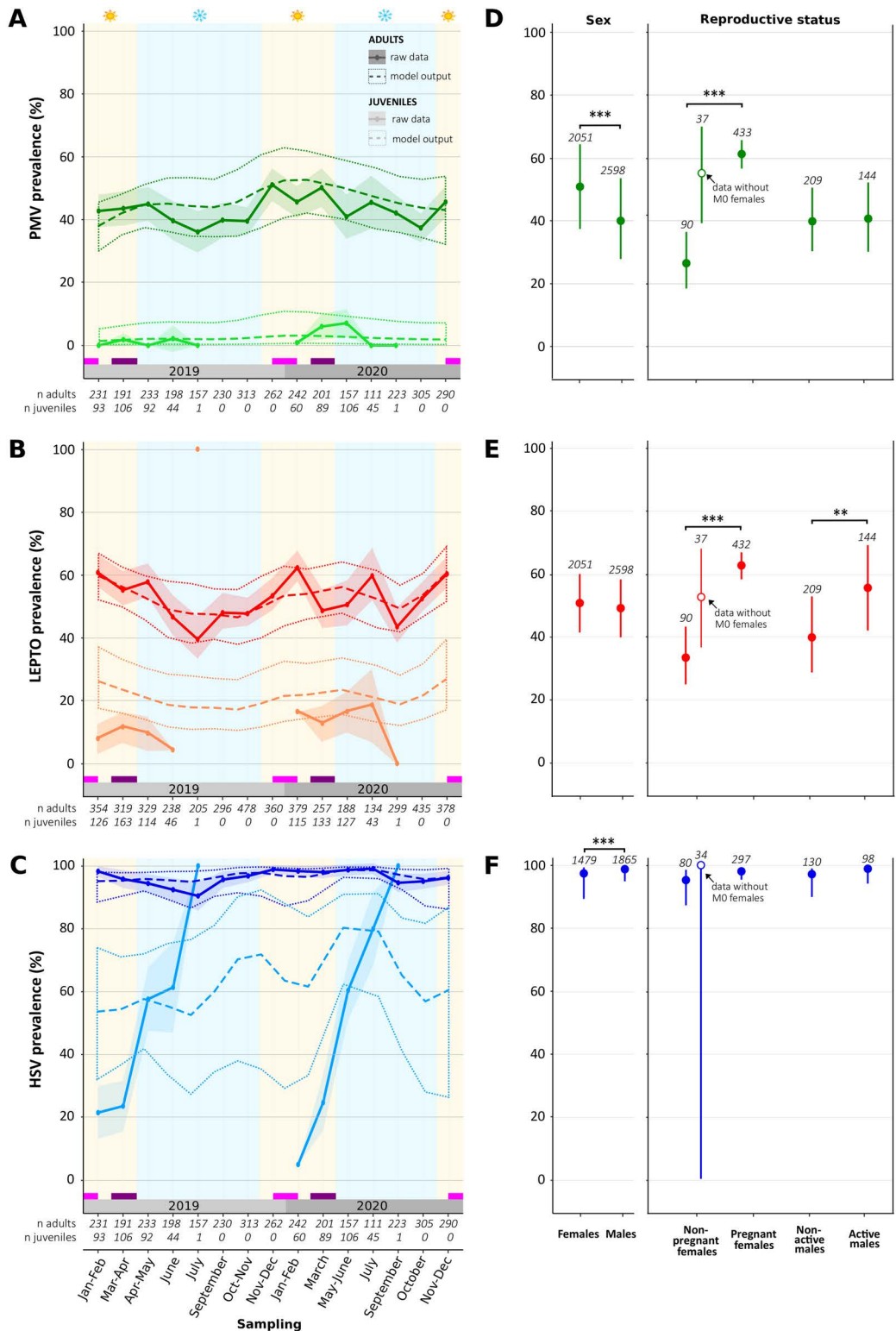

**Fig 3. Age and reproduction periods drive paramyxovirus (PMV), *Leptospira* bacteria (LEPTO) and herpesvirus (HSV) shedding dynamics in *M. francoismoutoui*.** (A-C) The observed prevalence (raw data) is represented by solid lines, with by 95% confidence intervals shown as shaded areas. Predicted prevalence and binomial sampling errors (model output) are depicted by dotted lines. Pink and purple blocks at the bottom represent the parturition and the mating periods, respectively. Austral summer and winter are indicated in yellow and blue, respectively. (D-F) Predicted prevalence

(and binomial sampling errors) in adults according to their sex and reproductive status (active or not). Significant effects of GAMs (S1 and S3 Tables) are indicated: *** $P < 0.001$ and ** $P < 0.01$.

M14: $z = -10.67$, $P < 2^{-16}$; model M15: $z = -7.39$, $P = 1.43^{-13}$; model M16: $z = -4.69$, $P = 2.68^{-06}$; S3 Table), especially in adult females (Tukey's post hoc test, model M13: $t = 5.04$, $P < 0.001$; model M15: $t = 4.26$, $P = 0.001$). Finally, pregnancy increased the probability of dual and triple shedding in females (model M17: $z = 4.01$, $P = 6.15^{-05}$; model M18: $z = 4.93$, $P = 8.05^{-07}$; model M19: $z = 5.07$, $P = 3.97^{-07}$; model M20: $z = 3.79$, $P = 1.5^{-04}$; S4 Table), but when non-pregnant females without visible nipples (M0) were excluded, this effect was no longer observed (S4 Table). In males, reproductively active individuals had a higher probability of simultaneously shedding LEPTO and HSV compared to the non-reproductive ones (model M22: $z = 2.66$; $P = 0.008$; S4 Table).

### Within-host dynamics of shedding

All captured bats were marked (by tattooing on the wing) allowing to collect shedding data from recaptured bats, and enable comparison of results from samples collected on initial capture and on recaptures. 351 bats were captured more than once, from which 407 and 393 samples were tested for PMV and LEPTO respectively. We also tested 86 saliva samples from 31 recaptured bats (from 2 to 5 capture events) for the presence of HSV. All bats initially testing HSV-positive (n = 29) remained positive over time (between 25 and 405 days between recapture events), while the only two initially-negative bats were found to be positive at their first recapture. Thus, within-host analyses using multinomial logistic regressions of shedding status transitions were only performed on PMV and LEPTO data. Recapture intervals ranged from 25 to 719 days (S3A Fig), during which we defined four transition classes: negative to negative (no transition), negative to positive, positive to negative, and positive to positive (no transition). A significant proportion of recaptured bats maintained their initial shedding status over time. Specifically, 50.74% (PMV) and 40.04% (LEPTO) of recaptured bats remained negative, while a smaller yet notable fraction (PMV: 25.69%, LEPTO: 34.79%) consistently remained positive. Notably, among initially-positive bats, approximately 75% remained positive over time, though this percentage slightly declined for PMV when bats were recaptured after 12 months (Fig 5A). Examples of bats that remained positive throughout multiple recapture events, both within the same year (short-time intervals) and across different years (long-time intervals) are illustrated in S4 Fig.

The probability of the four shedding status transitions was significantly dependent on the time interval between recaptures for LEPTO (model M25, S5 Table) (Fig 5B), but not for PMV (model M26, S5 Table) (Fig 5C). Indeed, the probability of shedding LEPTO (either negative to positive, or positive to positive) increased with time interval, while the probability of stopping shedding LEPTO (positive to negative) remained low and stable (Fig 5B). We also found a significant effect of changing reproductive status on the probability of changing both PMV and LEPTO shedding status (models M5 and M26, S5 Table). Indeed, bats that remained non-reproductively active (defined as adult females there were neither pregnant nor post-lactating, and adult males without large testes) over recapture events (reproduction transition: 0/0), also mainly remained LEPTO- and PMV-negative (shedding transition: 0/0) (Fig 5D and 5E). Similarly, most bats successively captured when reproductively active (reproduction transition: 1/1) remained LEPTO- and PMV-positive (shedding transition: 1/1).

## Discussion

Using an original island-endemic bat system, we demonstrated the widespread and all-year-long co-shedding of two viral and one bacterial agents in Reunion free-tailed bat populations. This confirms the circulation of PMV and LEPTO [10,44] and describes for the first time HSV in this bat species. We revealed high global prevalence of PMV (37%) and LEPTO (46%), and extremely high prevalence of HSV (87%), which contrasts with certain other bat-borne infectious

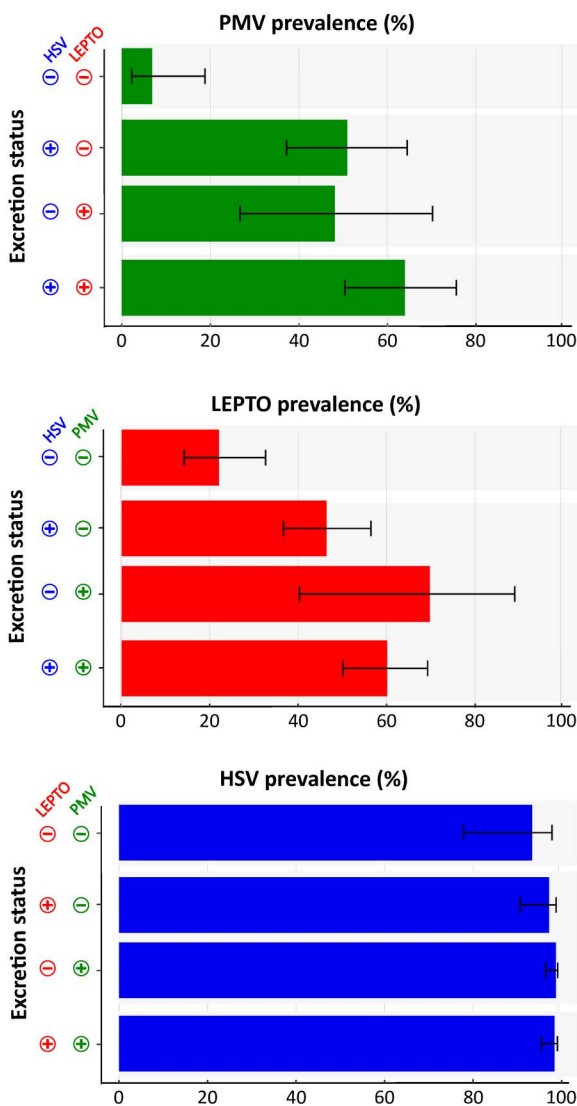

**Fig 4. Shedding probability increases with dual and triple infections in *M. francoismoutoui*.** Prevalence estimates were derived from the predicted probabilities generated by GAMs M1-M3. Shedding status is coded with 0 for non-shedding bats and 1 for bats shedding infectious agent(s). Black bars indicate the 95% confidence intervals.

agents, such as Henipaviruses, Filoviruses, and Lyssaviruses, where viral shedding is infrequently detected (e.g., [6,47]). Our nearly complete HSV prevalence aligns with the consistently high HSV rates observed in other bat species, such as in the Mexican free-tailed bat *Tadarida brasiliensis* [48] and in the common vampire bat *Desmodus rotundus* [14,49]. High shedding prevalence in Reunion free-tailed bats could be explained by a strong gregarious behaviour of these bats, which can aggregate up to 1500 adult individuals per m² [50]. The proximity of individuals within roosts can facilitate transmission through bat fluids (saliva, urine), in which case, roost size is often assumed to positively correlate with prevalence [51–53]. However, we did not find such a correlation, suggesting that roost size may not play a major role in the shedding of paramyxoviruses, *Leptospira* and herpesviruses in Reunion free-tailed bats. This aligns with previous findings on coronavirus in these bats, which demonstrated similar dynamics in two maternity colonies

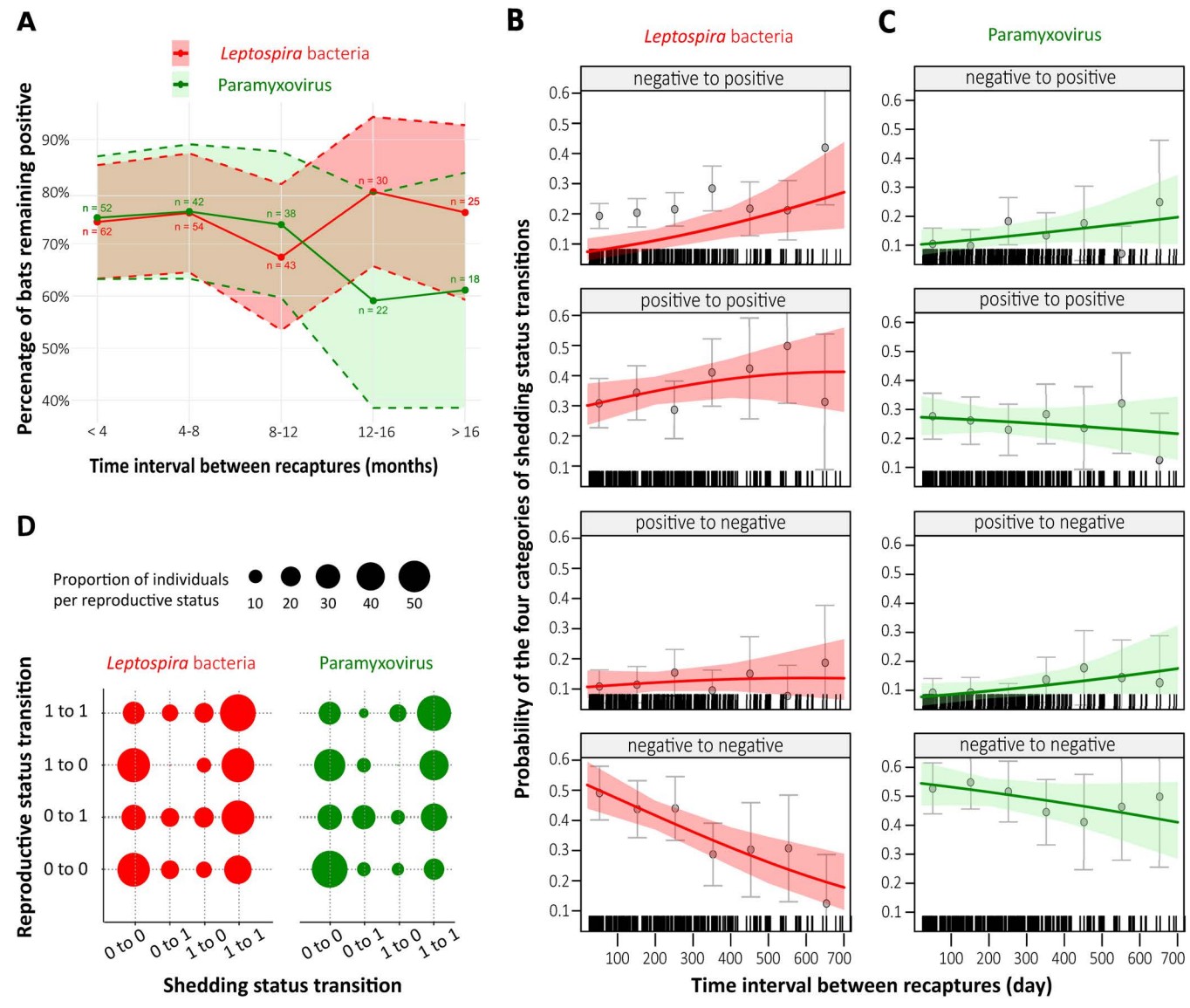

**Fig 5. Shedding histories in recaptured *M. francoismoutoui* bats.** (A) Percentage of bats remaining *Leptospira* and paramyxovirus positive through time. (B-C) Probability of the four categories of shedding status transitions over time, for (B) *Leptospira* and (C) paramyxovirus. Predicted values (output of the multinomial models) and 95% confidence intervals are depicted with colored lines and shaded areas. Raw proportions (and 95% CI) are represented with grey circles and bars for each interval of 100 days. (D) Proportion of recaptured bats according to changes in shedding status and reproductive status. Shedding status is coded with 0 for non-shedding and 1 for shedding bats. Reproductive status is coded with 0 for non-active and 1 for active bats.

with markedly different roost sizes [10]. Moreover, we observed a degree of spatial synchrony between roosts, with a common prevalence trend characterized by consistently high prevalence throughout the year, and an increase during the summer months. These correlated shedding dynamics were observed even between geographically distant roosts. The high prevalence in our current dataset, as well as the low variability across sampling periods, may have obscured a higher signal of detectable synchrony. Extending the time series could provide the necessary variation and statistical power to capture any underlying synchronous trends across roosts and confirm our initial conclusions. Although frequent intra- and inter-seasonal movements is suspected in Reunion free-tailed bats [12,45], potentially contributing to

pathogen spread and spatial synchrony, our results suggest that transient epidemics, marked by waves of bat-to-bat transmission between roosts, are not the primary mechanism driving global PMV, LEPTO and HSV shedding dynamics [8,51,54]. Instead, intrinsic biological cycles, which are influenced by individual (reproductive status, age, and immune function) and environmental factors (e.g., food availability, temperature) could create synchronized shedding patterns among roosts [5,51].

Several studies have reported that age structure within bat populations is a critical determinant of infection dynamics [5]. For coronaviruses, studies have documented higher infection rates among juveniles, who become rapidly infected shortly after birth [9,10,22]. In our study, the rapid increase in HSV infections (60% of infection in 4-months old juveniles) suggests a similar rapid waning of maternal immunity. In contrast, juveniles showed relatively low infection rates for PMV and LEPTO during their first six months of life. Additionally, observations of females with non-developing nipples, which were associated with low infection rates (S1 Fig), suggest that older juveniles may remain protected against PMV and LEPTO for an extended period. While we could not confirm this timing explicitly because of the incertitude in aging old juveniles, this is consistent with the longer maternal immunity reported in field and modelling studies for Lagos bat virus, henipaviruses and Marburg virus (e.g., [38, 55–58]). Further serological investigations are thus required to determine the duration of maternal antibodies in this bat species.

Behavioral and physiological traits are two mechanisms that can produce sex-biased prevalence in several wildlife species [59,60]. Here, adult males were more frequently shedding HSV than females, as previously reported in rodents [61] or for other DNA bat viruses such as adenoviruses [62]. In contrary, adult females were more frequently shedding PMV. Studies on HSV prevalence in bats are limited and report contrasting results, with no sex difference in *D. rotundus* [14], while Sjodin et al., [63] suggest that either males or females can be more infected, depending of the herpesvirus strain. Male-biased infection may arise from sex-specific social behaviour like dominance or higher aggressiveness (physical injuries, i.e., bite) [64,65], that could enhance transmission in males [61,66]. Sex hormones can also strongly affect host susceptibility, through the modulation of immunity [67]. Testosterone may suppress immune responses in males, increasing susceptibility to pathogen infection [68]. The inclusion of hormone and immunity measures should thus be considered in future infection studies to test this hypothesis in bats.

At the individual level, chronic or episodic shedding from persistently infected bats can lead to high prevalence across roosts and time [4]. However, deciphering between the different within-host processes that may lead to such persistence can be challenging. For HSV, in addition to an extreme high prevalence, our recapture data showed that all initially HSV-positive bats were still shedding HSV in saliva over recaptures, supporting indeed lifelong HSV infection in Reunion free-tailed bats. HSV are known to establish latent infection in humans [69] and rodents [61], and the same mechanism is also suspected in bats [70,71]. Interestingly, when modeling the infection *in D. rotundus* with a betaherpesvirus [14], authors showed that the length of the active period (when shedding occurs) appears to be longer than the latency state (when shedding stops) [71]. The same mechanism of latency and long phase of active shedding could explain our result, as suggested by previous modeling work, although there is no empirical evidence to support this result. Moreover, given that all individuals were shedding upon recapture, continuous shedding from persistent infections could also represents a plausible scenario. The recent deep sequencing of HSV-positive samples collected over three consecutive years—including some samples analyzed in this study—identified co-infections of Reunion free-tailed bats with multiple strains spanning three HSV subfamilies (alpha, beta, and gamma) [72]. These findings also indicated a low probability of strain loss over time, supporting within-host persistence. To distinguish between within-host processes contributing to HSV persistence in Reunion free-tailed bats, further sequencing efforts will be necessary to uncover potential subfamily- or strain-specific dynamics [49]), which may be masked within the global HSV shedding prevalence observed in the present study. This has, for example, been recently shown for distinct bat-borne coronavirus and paramyxovirus lineages [2,24,73].

For PMV and LEPTO, our extensive dataset of recaptured bats revealed that approximately 75% of initially positive individuals continued to shed viral RNA and bacterial DNA in their urine over extended periods. Additionally, examples of bats remaining positive across multiple recapture events further support the hypothesis that persistent infections contribute to the observed temporal dynamics of PMV and LEPTO. However, we also observed transitions from positive to negative shedding for PMV and LEPTO, indicating that cycles of clearance followed by reinfection may occur within the bat population. While continuous shedding is supported by the data, the possibility of recrudescence from latent infections cannot be ruled out and may also explain the results. Persistent LEPTO infection in bats is not surprising as this bacterial infection is often considered chronic in other reservoir hosts, because the bacteria form protective biofilms in renal tubules [74]. Experimental infections have shown that *Leptospira interrogans* can stably colonize the kidneys of rats and mice over their lifespan, with intermittent shedding of the bacteria that can last for at least 3 months [75,76]. Our data showed that when individuals were shedding LEPTO upon their first capture, the probability of stopping shedding was low and stable over time, which is in keeping with an intermittent shedding. On contrary, mechanisms underlying within-host PMV dynamics are still debated [4,77]. Several field and lab evidence support that henipavirus, for example, can persist within individual bats, with long-term shedding occurring either continuously or intermittently through recrudescence [8,38,78–81]. A modelling study also suggests that latent henipavirus infections can occur alongside occasional viral clearance, followed by temporary immunity and reinfection [77]. This same study supports that bats are much more likely to develop a latent infection than clearing it. Finally, it has been hypothesized that a small portion of super-long-shedder individuals with a long infectious period could maintain Hendra virus in bat populations [82]. Distinguishing between these different infection processes is complex. Because we only measured the presence of viral and bacterial nucleic acids in our study, a first step will be to assess the infectiousness of excreted samples to finely validate the role of long-term continuous and intermittent shedding in transmission [51]. Secondly, as we previously showed the presence of several paramyxovirus variants in Reunion free-tailed bats, but only one for *Leptospira* [44], further sequencing of positive samples from this study will be necessary to assess temporal dynamics at a finer taxonomic scale, and provide key information on viral and bacterial evolution within the bat host [4]. Together with further development of epidemiological models [83], such genetic data will help clarify the mechanisms shaping the observed shedding dynamics (e.g., [7,8,71,77]).

In persistent infections, the reactivation process is often linked to several stressors that may affect host immunity [84,85]. Reproduction, by imposing immunological challenges, is recognized as a major stressor that could lead to the reactivation of viral and bacterial shedding [86]. Such processes are expected to be particularly pronounced in female mammals, for whom reproduction requires large and prolonged energetic investments [87,88]. Although less documented, male reproduction trade-offs have also been suggested in bats [89] and could be mediated by reduced immune function during spermatogenesis, a period of high energy demand and elevated testosterone [90–92]. Our results suggest that pregnancy increases the probability of PMV and LEPTO shedding, as well as the likelihood of dual and triple shedding in Reunion free-tailed bats. Moreover, our study suggested that, during the mating period, reproductively active males had a higher probability of shedding LEPTO (and LEPTO-HSV), as compared to non-reproductive males. Individual recapture histories also suggest that changes in reproductive status influence PMV and LEPTO shedding. The lack of effect for HSV is likely due to its high prevalence, which may obscure any potential reproductive influence, as also observed in HSV-infected *Rousettus aegyptiacus* and *Miniopterus natalensis* [93]. We thus hypothesize that shedding could be reactivated during pregnancy and mating periods in Reunion free-tailed bats, as reported in other bat species [10,22,23,44,93,94]. Such a process, besides conferring protection to newborns through the transfer of protective maternal antibodies, could benefit the adults through a boost of immunity and could therefore be critical in the maintenance of long-term immunity [95].

While reproduction is often recognized as a key factor shaping infection dynamics—and several of our findings align with this pattern—an age-related effect could equally explain the observed seasonal variations in pathogen shedding. Indeed, distinguishing adult bats from old juveniles (> 6 months-old) in the field, based on morphology alone, was

particularly challenging in our study. This difficulty may have introduced bias into our observations of increased prevalence at the end of winter and the subsequent peak during gestation (October–December). Specifically, these trends could partly reflect the progressive infection of older juveniles misclassified as adults. Similarly, during the mating period (March–April), adult male groups may include some misclassified year-old juveniles (approximately 15 months old) with lower infection rates. Finally, immigration of infected bats may also play a role in transmission during late pregnancy especially, as massive aggregations of bats occur at this time of the year with possible horizontal transmission to non-immune bats [12].

Our investigation targeting multiple simultaneous infectious agents revealed a high co-shedding of viruses and bacteria within the bat population, with 59% of bats shedding at least two infectious agents. Bats shedding one virus or bacteria had significantly more chance of shedding another one, with a strong association between viral infections in particular. This corroborates previous results suggesting positive interactions that could facilitate shedding of paramyxoviruses [2,44] and coronaviruses [18,26]. In the study of Davy et al. [18], the stress of fungal infection by *Pseudogymnoascus destructans* led to a reactivation of coronavirus replication in the little brown bat *Myotis lucifugus*, and co-infection even increased the severity of White Nose Syndrome, illustrating how multiple infections can interact to affect host health. Interestingly, our study also found that dual infection also increased the likelihood of shedding a third infectious agent, although combined effect of two infections on the third was lower than expected. These interactions between infectious agents could modulate host susceptibility in various ways, with both antagonistic and mutualistic effects, depending on the community of infectious agents considered [30]. In our previous work, we observed that coinfection with paramyxoviruses, coronaviruses and astroviruses was a very rare event (1% of tested samples) in Reunion free-tailed bats and no significant interactions were detected between these viruses [10]. This is probably explained by the very low prevalence of astroviruses, and the asynchronous timing of juvenile coronavirus infection, which occurs between February and April [22], when they are still low infected by PMV. These studies illustrate how considering simultaneously multiple infectious agents, although complex, is important to better understand drivers of infection in bats [19,96,97]. Disruption of balanced host-pathogen relationships due to stressors, such as habitat loss or co-infections, may increase pathogen shedding, and enhance the likelihood of spillovers [3]. Further research exploring the dynamics of multi-infections in the context of increasing urbanization, particularly in species like the urban-dwelling Reunion free-tailed bat, will help in understanding how human-mediated changes in bat ecology may affect natural shedding dynamics and ultimately the probability of spillovers [98].

## Materials and methods

### Ethics statement

Bats were handled using personal protective equipment regularly changed and all the equipment was disinfected between sites (more details in [12]). Bat capture and manipulation protocols were evaluated by the ethic committee of Reunion Island (CYROI, n°114), and approved by the Minister of Higher Education, Research and Innovation (APA FIS#10140–2017030119531267), and conducted under a permit delivered by the Direction de l'Environnement, de l'Aménagement et du Logement (DEAL) of Reunion Island (DEAL/SEB/UBIO/2018-09).

### Study sites and field sampling

We monitored and sampled 17 roosts (coded with a 3-letter code) from January 2019 to December 2020 (24 months), all over Reunion Island (Fig 1). Among the 17 studied roosts, 12 were monitored across a maximum of 15 successive "sampling periods" occurring every 3–8 weeks, each sampling period lasting for 3–4 weeks to be able to sample multiple roosts (Table 1) (S6 Table). An opportunistic sampling was performed for five additional roosts, which were thus monitored only once during the studied period. To investigate within-host dynamics, we used data from recaptured bats, and thus also included previously collected samples from 2018 (referred to as the first capture samples).

Bat monitoring includes bat capture, biological sampling and roost size estimation and is fully described in Aguillon et al. [12]. Captures were carried out using harp traps (Faunatech Ausbat) and Japanese monofilament mist nets (Ecotone) during dusk emergence, with a maximum of 60 individuals captured per night. Bats were hydrated immediately after the capture and kept individually in a clean individual bag hang close to a warm water bottle. For each bat, we measured the forearm length and mass, and determined the sex, age, and reproductive status. Age was established by examining the epiphysis fusion in finger articulations that are not welded for juveniles which was easily identifiable from January to July, while some older juveniles might have been classified as adults in the following months. In females, we recorded the development of nipples as M0 for non-visible nipples, M1 for visible nipples, M2 inflated nipples (lactating), and M3 (visible nipples with regrowth of hair, post-lactating). Active reproductive status was reported when females were pregnant (also verified by palpation of the abdomen), lactating and post-lactating, and in males when they had large testes (see [12] for more details). A sterile swab (Puritan Medical Products, USA) was carefully inserted in the corner of the lips to sample saliva and then placed in 250uL of MEM (Minimum Essential Medium Eagle). Using a pipette and a sterile tip, urine droplets were collected in the waterproof bag or directly at the urethral opening when handling bats, and then placed in 50 μL of MEM. All samples were stored in a cooled box in the field before being transferred at -80°C at the laboratory the same night. Finally, each bat was tattooed on the right propatagium with an individual alphanumeric code [99] and released at the capture site.

## Molecular detection of infectious agents

Urine and saliva samples were processed using the Cador Pathogen 96 Qiacube HT kit (Qiagen, Hilden, Germany), using 25 μL and 100 μL of samples, respectively, mixed with 175 μL and 100 μL of VXL buffer for the lysis step. Total nucleic acids were extracted using an automated extractor Qiacube with small modifications of the Q Protocol using 350 μL of ACB, 100 μL of AVE, and 30 μL of TopElute. A reverse transcription was then performed on urine extracts [100], followed by the screening of paramyxoviruses and *Leptospira* bacteria, using respectively a semi-nested PCR targeting the RNA polymerase gene of Rubula-, Morbilli- and Henipa-virus genera, and a real-time PCR amplifying a fragment of the 16S rDNA gene of several pathogenic *Leptospira* species [44,101,102]. For paramyxoviruses, amplicons of the first PCR were diluted to 1:10, and 2 μL was then used in the semi-nested PCR.

For saliva extracts, a fragment of herpesvirus DNA polymerase was amplified using a nested PCR targeting a broad spectrum of herpesviruses (including α, β and γ sub-families). During the first PCR, 2 μL of DNA was amplified in a 25 μL mixture containing 12.5 μL of GoTaq Green Master Mix 2X (Promega, Madison, Wisconsin, United States), 2.5 μL of each primer at 10 μM DFA (5'-GAYTTYGCNAGYYTNTAYCC-3') and KG1 (5'-GTCTTGCTCACCAGNTCNACNCCYTT-3') [103]. We used the following thermal conditions: 95°C for 2 min, and 45 cycles at 94°C for 30 sec, 46°C for 1 min and 72°C for 1 min, and a final step at 72°C for 7 min. During the nested PCR, 5 μL of amplificon was amplified in a 25 μL mixture containing 12.5 μL of GoTaq Green Master Mix 2X (Promega, Madison, Wisconsin, United States), 2.5 μL of each primer at 10 μM TGV (5'-TGTAACTCGGTGTAYGGNTTYACNGGNGT-3') and IYG (5'-CACAGAGTCCGTRTCNCCRTADAT-3') [103]. We used the following thermal conditions: 95°C for 2 min, and 35 cycles at 94°C for 30 sec, 46°C for 30 sec and 72°C for 30 sec, and a final step at 72°C for 7 min. PCR products were revealed using electrophoresis with 2% agarose gel stained with 2% GelRed (Biotium, Fremont, USA).

## Statistical analyses

We quantified the spatial (between roosts) synchrony in prevalence by calculating Pearson's cross-correlations between the time series, for seven roosts (ESA, MON, PSR, RBL, RPQ, TGI and VSP) that have complete temporal data (n = 15 sampling periods). The relation between synchrony and geographical distance was then modelled using the non-parametric spatial covariance function Sncf in the *ncf* package [104], as in Paez [105]. For that, we visually represented synchrony using spline correlograms, using a bootstrap permutation test (n = 500 iterations) to generate 95%

confidence intervals for the covariance function. Roosts sampled at least at eight different periods (n = 10) were used to fit the non-parametric correlation function, and missing data were handled via pairwise deletion of missing values for each pair of time series. Next, prevalence trends and synchrony were characterized using dynamic factor analysis (DFA) within the MARSS package [106]. Common trends were modeled by random walk, while loadings were estimated for each roost to describe the relationship between roost-specific time series and the common trend [107]. Time series were standardized to Z-scores (mean = 0, sd = 1) and modeled with a single common trend and an identity variance-covariance matrix.

To evaluate spatio-temporal shedding dynamics, we fit binomial generalized additive mixed models (GAM, [108]) via restricted maximum likelihood (REML) using the *mgcv* package (S1 and S3 Tables). We used penalized smoother for the temporal trend variable and controlled that the effective degrees of freedom (EDF) were higher than 1 (non-linear effect). The number of basis functions (k) and smooth parameters (alpha) were adjusted in each model using *gam.check* from the *mgcv* package to avoid overfitting and check for model convergence. Temporal trends were incorporated by assigning a numerical value to each sampling period, ranging from 1 (January-February) to 16 (November-December, S6 Table). Due to the COVID-19 crisis and the resulting lockdown in France, we did not monitor roosts in April 2020, and do not present data at that time. To properly account for the temporal effect, we decided to code the May-June 2020 sampling period as the middle of the notation used for March (coded as 10) and July 2020 (coded as 13, S6 Table). Final models for single, dual and triple shedding included the sampling period as smooth function, the roost size, age and sex (and their interaction) as linear function (S1 and S3 Tables). The significance of each modality of age/sex interactions was tested using a Tukey's post hoc test. For single shedding models, we also included the two other infectious agents, and their interaction, as a linear function. The influence of sexual spatial segregation patterns in adults on the prevalence was tested using the Sexual Segregation and Aggregation Statistic (SSAS, [109]). This statistic compares observed aggregation and segregation patterns to a random association of males and females among roosts. We calculated SSAS values for each sampling period, based on capture data reflecting the spatial distribution of individuals within roosts. The SSAS was included in models, as a smooth function (details in S1 and S3 Tables). We also assessed the difference in bacterial load of *Leptospira* between juveniles and adults by including age as a predictor in a GLM, using the cycle threshold as a proxy for bacterial load.

To assess the effect of reproductive status on shedding prevalence and bacterial load (using cycle thresholds as a proxy), we focused on adult bats at two specific periods of the year: firstly, in adult females during the pregnancy period (November-December) to compare pregnant and non-pregnant adult females. We used a separate GLM on a subset of data, for each sex, and included reproductive status as a linear effect (S2 and S4 Tables). During the pregnancy period, some non-pregnant females were classified as adults, based on fused epiphyses, but lacked developed nipples (M0), which could represent truly sexually mature adults that have not yet reproduced, as well as misclassified sexually immature juveniles (about 11 months-old). To avoid including these misclassified juveniles in the analysis, we ran the model both with and without the non-pregnant M0 females. For males, our focus was on adult males during the mating period (March-April), comparing reproductively active with non-reproductively active ones. It is important to note that at this time of the year, males classified as adults may also include some misclassified old juveniles (about 15 months-old) that could not be morphologically distinguished [12].

To investigate within-host shedding dynamics, we tested the probability of changing shedding status through time, by fitting a multinomial logistic regression using the *multinom* function in the *nnet* package [110] (S5 Table). Because not enough HSV-negative saliva samples were available to analyse transitions in HSV shedding status, we only modelled transitions in shedding status for PMV and LEPTO. Changing shedding status was used as a nominal dependent variable with four levels (negative to negative, negative to positive, positive to negative, and positive to positive), with the following covariates: recapture interval (in days), sex, and reproductive status transition (active to active, non-active to active, active to non-active and non-active to non-active). Age classes were not included in the models because of the limited number of juveniles compared to adults ($n_{JUV}$ = 29, $n_{AD}$ = 497). This analysis used the non-changing shedding status, negative

to negative, as a baseline outcome against which to compare other shedding status transitions. To better visualize the probable continuous shedding over time, we calculated and plotted the percentage of bats remaining positive for PMV and LEPTO across different time intervals: from < 4 months to > 16 months.

All analyses were performed in R.4.4.2 [111] using packages *dplyr*, *emmeans*, *lme4*, *ggeffects*, *ggplot2*, gridExtra, MARSS, *mgcv, ncf, reshape2, synchrony,* and *vegan*.

## Supporting information

**S1 Table. Summary of the statistical models (models M1 to M4) used to analyse single shedding dynamics in *M. francoismoutoui*.** Significant variables are in bold and the asterisk represents the interaction between two variables. All GAMs were fitted with a binomial distribution. Model M4 (GLM) was fitted with a Gaussian distribution, and its performance was compared to a null model using AIC criterion. The percentage of deviance explained was calculated by comparing full model with null model. PMV: Paramyxovirus, LEPTO: *Leptospira* bacteria, HSV: Herpesvirus.
(DOCX)

**S2 Table. Summary of the statistical models (models M5 to M12) used to analyse the mono-excretion dynamics in *M. francoismoutoui* during pregnancy and mating periods, specifically.** GLMs were fitted with a binomial distribution, except for Ct LEPTO variable fitted with gaussian distribution. Final models (in bold) were selected by comparing full model with null model and using best AIC criterion (when ΔAIC > 2). The percentage of deviance explained was calculated by comparing full model with null model. PMV: Paramyxovirus, LEPTO: *Leptospira* bacteria, HSV: Herpesvirus. M0: female with no visible nipples.
(DOCX)

**S3 Table. Summary of the statistical models (models M13 to M16) used to analyse dual and triple shedding dynamics in *M. francoismoutoui*.** Significant variables are in bold and the asterisk represents the interaction between two variables. The percentage of deviance explained was calculated by comparing full model with null model. All GAMs were fitted with a binomial distribution. PMV: Paramyxovirus, LEPTO: *Leptospira* bacteria, HSV: Herpesvirus.
(DOCX)

**S4 Table. Summary of the statistical models (models M17 to M24) used to analyse dual and triple shedding dynamics in *M. francoismoutoui* during pregnancy and mating periods specifically.** GLMs were fitted with a binomial distribution. Final models (in bold) were selected by comparing full model with null model and using best AIC criterion (when ΔAIC > 2). The percentage of deviance explained was calculated by comparing full model with null model. PMV: Paramyxovirus, LEPTO: *Leptospira* bacteria, HSV: Herpesvirus. M0: female with no visible nipples.
(DOCX)

**S5 Table. Summary of the statistical models (models M25 and M26) used to analyze recapture data in *M. francoismoutoui*.** Transitions in Herpesvirus shedding status was not analysed because not enough negative saliva samples were available. Interval: time interval (days) between recaptures. Repro: reproductive status transition (active to active, non-active to active, active to non-active and non-active to non-active). PMV: Paramyxovirus, LEPTO: *Leptospira* bacteria.
(DOCX)

**S6 Table. Characteristics and coding of sampling periods used in GAMs.**
(DOCX)

**S1 Fig. Temporal variation of paramyxovirus (PMV), *Leptospira* (LEPTO) and herpesvirus (HSV) prevalence in *M. francoismoutoui* females, according to age and reproductive characteristics (pregnancy and visibility of nipples).**

Note that in the green group, bats can include both adult bats that were not sexually mature yet, as well as misclassified juveniles.
(DOCX)

**S2 Fig. Venn diagram of *M. francoismoutoui* bats tested for the three infectious agents (n = 3784).** Observed proportions are shown with 95% confidence intervals (CI), and expected prevalence and 95% CI are indicated below in square brackets.
(DOCX)

**S3 Fig. Details of recaptured *M. francoismoutoui* bats.** (A) Distribution of time interval between recaptures. (B) Variation of time intervals across the four categories of shedding status transitions, for *Leptospira* and paramyxovirus. Shedding status is coded with 0 for non-shedding and 1 for shedding bats.
(DOCX)

**S4 Fig. Examples of recaptured bats with always-positive status for paramyxovirus and *Leptospira*.** The time interval (in days) between two recaptures is indicated above the arrow.
(DOCX)

## Acknowledgments

We are grateful to Eco-Med Océan Indien, Biotope, the DEER of Région Réunion (Direction de l'Exploitation et de l'Entretien des Routes), the DRT of Département Réunion (Direction des Routes et des Transports of the Department Reunion), and the City hall of Salazie for their help in identifying and accessing bat roosts. We are grateful to David Wilkinson for fruitful discussions. We also thank Clara Castex, Timothée Chenin, Jérémy Dubrulle, Marie Köster, and Guillaume Verchère for their assistance in the field.

## Author contributions

**Conceptualization:** Samantha Aguillon, Muriel Dietrich.

**Formal analysis:** Samantha Aguillon, Muriel Dietrich.

**Funding acquisition:** Pablo Tortosa, Muriel Dietrich.

**Investigation:** Samantha Aguillon, Magali Turpin, Gildas Le Minter, Camille Lebarbenchon, Axel O. G. Hoarau, Céline Toty, Avril Duchet, Léa Joffrin, Riana V. Ramanantsalama, Pablo Tortosa, Muriel Dietrich.

**Supervision:** Patrick Mavingui, Muriel Dietrich.

**Validation:** Muriel Dietrich.

**Visualization:** Samantha Aguillon.

**Writing – original draft:** Samantha Aguillon, Muriel Dietrich.

**Writing – review & editing:** Samantha Aguillon, Magali Turpin, Gildas Le Minter, Camille Lebarbenchon, Axel O. G. Hoarau, Céline Toty, Avril Duchet, Léa Joffrin, Riana V. Ramanantsalama, Pablo Tortosa, Patrick Mavingui, Muriel Dietrich.

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
