## [Decision Letter · Decision Letter 0]

Dear Dr Aguillon,

Thank you very much for submitting your manuscript "Role of individual and population heterogeneity in shaping dynamics of multi-pathogen excretion in an island endemic bat" for consideration at PLOS Pathogens. As with all papers reviewed by the journal, your manuscript was reviewed by members of the editorial board and by several independent reviewers. In light of the reviews (below this email), we would like to invite the resubmission of a significantly-revised version that takes into account the reviewers' comments.

We cannot make any decision about publication until we have seen the revised manuscript and your response to the reviewers' comments. Your revised manuscript is also likely to be sent to reviewers for further evaluation.

Sincerely,

Katia Koelle

Guest Editor

PLOS Pathogens

Ronald Swanstrom

Section Editor

PLOS Pathogens

Michael Malim

Editor-in-Chief

PLOS Pathogens

orcid.org/0000-0002-7699-2064

Reviewer's Responses to Questions

**Part I - Summary**

Reviewer #1: Aguillon et al present an extensive longitudinal study on the simultaneous excretion of paramyxoviruses, herpesviruses and Leptospira bacteria in the Reunion free-tailed bat across 17 roosts over 24 months. The authors detect year-round excretion at high prevalence, with little temporal synchrony among roosts. Longitudinal studies of infectious agents in bats are still rare and so this study presents a valuable contribution. In particular, the multi-agent focus here is particularly rare, as is the high recapture rate of individuals. The latter is almost unheard of in bat studies. This allowed the detection of individuals that are infected across multiple sampling periods - interpreted here as persistent infections and cycles of clearance and reinfection. If this is indeed the case, it provides critical data on mechanisms of viral maintenance in bat populations - a major point of conjecture in this field.

One critical gap in the analyses is analyses and a description of whether the detections here represent multiple species, variants or strains of each of the viral families or the bacteria genus. The analyses implicitly imply that it is a single paramyxovirus species, a single herpesvirus species, and a single species of Leptospira, however I did not see this examined or even if the PCR was family-level or species specific. This should be clarified explicitly and given that the entire interpretations rest on this assessment, this would be imperative to be addressed by revisions.

Also, even the detections do represent single species within each family, it is a shame not to see any genetic analyses to confirm whether these repeated positive detections are persistent infections or re-infections. While persistent infections of herpesviruses (and I think Leptospira) are well known, evidence for the same for paramyxoviruses is more circumstantial. Inclusion of such data would be a huge boost to this study and the inferences that can be made. I would strongly encourage the authors to consider adding that work to this manuscript - with it, I think this paper could emerge as a foundational study for future work in the field.

Identifying the drivers of patterns in excretion remains challenging - here, it seemed difficult to ascertain age in some months, and the proportion of juveniles captured was very low (with misclassification as an identified issue). This is concerning given the results suggest that transmission is highly dynamic in this cohort - at least for HSV. Rigorous assessment of the statistical approaches is outside of my expertise, however I suspect that there could be some issues there - e.g with the low % of explained deviance in some of the GAMMS, and the absence of model comparisons e.g. through AIC. Other reviewers may be able to provide more specific feedback here.

The rates of co-excretion of pathogens is also exceptional. In the co-infection results section, I would have liked to see more attention to the nature of the bi-infecting agents e.g. bacteria-virus co-infections versus virus-virus co-infections, and whether the nature of the bi-infection affected the likelihood of tri-infection. I had trouble interpreting the figure. I think this is an incredible dataset that may warrant more sophisticated co-occurrence analyses that takes dependencies into account.

Overall, I'd like to encourage the authors - The major contributions of this study are the longitudinal dataset, high rates of recapture and repeated positivity, and the high rates of co-infection. These factors provide the opportunity for major insights into the mechanisms driving pathogen transmission in bat populations. However, I feel that the current analyses and clarity of writing are not rigorously assessing and clearly communicating the value of this dataset and these results. I would like to see a revision that addresses these points.

Reviewer #2: This manuscript examines patterns of pathogen shedding for three pathogens (Herpes and Paramyxovirus viruses and Leptospirosis bacteria) over space and time (12 roosts sampled ~15 times, and 5 sampled once) in of a single species of bats on an oceanic island. The dataset is moderately large (3700 samples overall, but only ~20/site-sampling period). This type of dataset is extremely rare and contrasts markedly with anecdotal sampling often carried out when sampling bats. Further, the study examines pathogen shedding (rather than serology) which is especially interesting since it is of more importance for transmission.

The broad patterns are very interesting – shedding of HSV is nearly complete (87% of all bats), whereas shedding of the other two pathogens is very high (30-50% of all bats). This contrasts strongly with some other bat pathogens in which shedding is rarely detected (e.g. Henipaviruses, Filoviruses, Lyssaviruses). Temporal patterns of shedding are only weakly interesting, likely due to the very short study period (24 months). The spatial patterns aren’t examined in any meaningful way except a poorly executed synchrony analysis (see below for details). Coinfection patterns are likely quite interesting, but the analyses and data presentation don’t allow the reader to assess the data properly. There were some differences in shedding with individual traits (e.g. sex, age, reproductive status), but some of these analyses were problematic and likely spurious (details below).

Overall, the study has a very nice data set, but the analyses were highly problematic (extensive details below) which makes the conclusions unsupported. If the authors revise the analyses extensively, this could be a very important contribution to our understanding of pathogen dynamics in bats.

**Part II – Major Issues: Key Experiments Required for Acceptance**

Reviewer #1: - sequencing to confirm viral/bacterial species and that there are in fact not multiple co-circulating paramyxoviruses, multiple co-circulating herpesviruses and multiple co-circulating Leptospira species

- to fully justify claims of persistent infections, sequencing of sufficient depth to demonstrate identical strains within the same individual over time, subject to minor within-host evolution

Reviewer #2: Major statistical issues:

- The paper doesn’t actually show any of the statistical models including coefficients, SEs, P-values, etc. Add a supplemental table with full details for each of the models M1-M3, and M10-M13, and ideally for all the other models as well. The table should indicate the details of the statistical model (what distribution was used for the response? What link function?), the full results (for each predictor include the coeff, SE, Z/t value and exact P-value (NOT <0.05)). Indicate the variance/deviance explained of the model and compare this to a null model that includes the same random effects so it’s clear how much additional variance/deviance the fixed effects are adding.

- The analyses are set up in a way that makes it unlikely they will detect some effects even if they are present and strong (e.g. colony size – see below).

- They analyses include random effects for things that shouldn’t be random (e.g. there are only 2 years of data and yet “year” is a random effect; one can’t/shouldn’t estimate a variance with 2 data points).

- The fitted models have many redundant terms – e.g. sampling period and julian date (the latter as a random effect which means it will be treated as a factor, not a continuous variable). These are both explaining the same thing (temporal patterns) and the inclusion of both obscures the actual explanatory power of each one.

- The paper argues there was no synchrony in pathogen dynamics among roosts, when the study length is simply far too short (2 yrs) to detect it. Further there was actually evidence of synchrony for one of the 3 pathogens despite the very short study duration suggesting synchrony is likely much higher and would be easily detectable across reasonable time periods. Further the analysis the authors use, a rank test, is a terrible way to analyze this data because it has very little power.

- Overall, it’s quite disappointing that the authors don’t take more advantage of the data and fit mechanistic models (such as those in Brook et al 2019 J. Animal Ecology, Epstein et al 2020 PNAS) so they can learn much more about the biology underlying the data. The GAMMS used here are a very poor substitute.

Data presentation

The data presentation is generally poor. None of the figures that show the shedding prevalence data (e.g. Fig 2A, Fig S2) show the uncertainty in the prevalence. The figures show the model uncertainty but that is quite different from the binomial uncertainty for a given sample. Sample sizes are never shown on any of the figures. Add them to the figures.

Language issues

The paper needs to be read by a native speaker. It contained dozens/hundreds of grammatical issues, misuses of words, and was quite difficult to understand.

Logical/conceptual issues

The response variable the study measured – pathogen shedding – is not the same thing as infection, but in many places the authors appeared to treat the two as the same. They are not the same and comparisons between studies that measured either infection or measures of past infection (e.g. serology) cannot be easily compared. One wouldn’t expect the same patterns over time or among individuals or ages.

Finally, there is a huge unanswered question that the authors strangely don’t address. They show patterns of pathogen shedding for adults and juveniles. For 2 of the 3 pathogens shedding prevalence in juveniles is a tiny fraction of that in adults. It’s not clear when juveniles get infected and start shedding at prevalences that match the adults. The gap between juveniles and adults likely explains several of the patterns the authors attribute to other factors (e.g. the effect of reproduction – see below). Most worrisome, it appears the authors weren’t able to age bats during the key part of infection and shedding (after 6 months of age), making the overall patterns potentially simply a result of including a variable number of old juveniles in different samples. Can the authors use the size measurements (e.g. forearm length, mass) plus recaptures to identify the ages of individuals more carefully?

**Part III – Minor Issues: Editorial and Data Presentation Modifications**

Reviewer #1: In the attached marked up PDF, I have highlighted some sentences where the meaning was not clear, and made some suggested minor changes to wording throughout. I have also underlined sections in where the phrasing wasn't clear and could be improved or the grammar was incorrect. General editing for clarity and grammar would be helpful, however overall, the ideas and intended meaning was mostly clear and the structure was logical. Apologies if some of my comments are addressed within the manuscript - hopefully revision of the text will help make these points more explicit and clear.

Some additional points are below (in addition to those in the PDF):

- The disparity between the number of juveniles caught and adults caught suggest to me that there is misclassification occurring. This is mentioned at one point, however I think that it it needs to be more clearly highlighted as a limitation to interpretation in the results and the discussion

- Further discussion of the consistency of the minimal detections in juveniles with previous studies in other bat species is warranted

- linking density with roost size is problematic. An initial assumption in then comparing this with prevalence (and making statements about density dependence) is that roost size correlates with density. Is there evidence that that is the case? ie. that the density of individuals per sq-m changes as the population grows? If these data are not available, then references to density should be removed, and just focus on roost size. Density and roost size are frequently conflated in viral transmission studies in bats.

- With the high prevalence rates and inability in telling apart persistent infection versus re-infection, I would have thought that it would be challenging to confidently assess synchrony and metapopulation dynamics. I think this needs to be toned down or further justified.

- I found the discussion around reactivation of PMV and Leptospira infections during pregnancy interesting, but unconvincing given the lack of seasonality in infections. I would like to see versions of figure 2B that incorporate time - i.e. can you compare pregnant versus non-pregnant females during the season when most female bats are mid pregnancy. Also, given the challenge in differentiating first year juvenile bats of adult size at this time, how well can this effect be separated from age? Again, sequencing of positive recaptured bats would be a huge contribution to these inferences.

- overall I felt that the discussion was too speculative. I would suggest focusing only on the more strongly supported results, aiming for an integration of the discussion points in a way that pulls together the most parsimonious hypotheses for the whole study. Currently, there are many disconnected mechanisms discussed to explain individual results, without a strong connected thread throughout the discussion

Reviewer #2: These are NOT minor comments. They are detailed comments pertaining to individual parts of the paper.

Detailed comments

L58-60 This sentence contradicts itself. Infection is either persistent or gets cleared. It can’t be both.

L62 Do you mean “recrudesce”?

L49-70 There are many other studies like this one but the authors make no effort in the abstract to acknowledge this work, build on it and indicate what new has been learned from this study. I’d suggest condensing the current abstract to 2/3 of it’s current length and adding both an introductory sentence that lays out what currently isn’t known and then 1-2 concluding sentences that indicate how the results from this study further our general knowledge. Otherwise. the work is only of limited interest outside of Reunion.

L75 How is it a “model”?

L79-80 See comment above. This is a self contradiction.

L82 Really? Most other studies suggest reducing spillover primarily involves reducing human-wildlife contact.

L85 There is no definitive evidence that SARS-CoV-2 is a bat virus. Revise this sentence to reflect this. There are many other pathogens where the reservoir hosts are known (e.g. SARS-CoV-1, Nipah virus, rabies virus, etc.). Use one of these instead.

L88 Ref 4 is a review and has no original data, so it’s not a great ref for this statement. There is no statistical analysis linking spillover to shedding in ref 2. Finally, the analyses in Ref 3 linking shedding to spillover are quite problematic – the data are pseudoreplicates and are analyzed a half dozen ways without any attempt to control for multiple comparisons. If there are other papers that show links between shedding and spillover, please cite them here instead of refs 2-4.

L92 There are several robust datasets that must be acknowledged here, including ref 11, as well as work by Heymann et al., Brooks et al JAE, etc.

L106 Add a key reference on coinfections: Graham 2008 PNAS www.pnas.org cgi doi 10.1073 pnas.0707221105

L115 Replace “flight” with “fly”

L117 do you mean “on islands”?

L188 Replace “help sample” with “enable sampling”

L112-132 This paragraph adds nothing to the paper and should be deleted

Figure 1 – why are you showing prevalence with the white portion of pie charts? This is counterintuitive and will confuse the reader. Please use the color for prevalence. Even better, use a bar chart rather than a pie chart. Pie charts are well known to be misleading for visualizing data. See, for example:

https://scc.ms.unimelb.edu.au/resources/data-visualisation-and-exploration/no_pie-charts

Fig 2. Are you really showing GAMMs? It looks like you are just connecting the prevalence with lines and not showing binomial sampling error for each point (nor sample size).

Fig 2. Please add letters to all panels of the graph, not groups of panels.

Fig 2. For the HSV prevalence panel (bottom right) there is so much whitespace that it’s not possible to see the differences (or lack thereof) among the groups. Can you change the y-axis limits to be 80 – 100?

Fig S1 legend Replace “is resumed” to “is shown”

Fig S1 If the goal is to examine synchrony in infection dynamics of individual pathogens among roosts then it would be better to make 3 graphs – one for each pathogen – and show all the roosts on that graph. It’s currently very difficult to see the synchrony or lack thereof in Fig S1.

L187 What is M1? You need to tell the reader what the M1, M2, etc. are since the results appear before the methods.

L202 Overall synchrony may be low, but it appears to be higher for Herpesvirus, with 4 of the roosts being synchronous, whereas only one pair of roosts was synchronous for paramyxo.

L208 Report the P-values. Don’t just say P<0.05. There is nothing magical about 0.05.

L208 and Results: Where are the details of the statistical analyses? You MUST provide a table of the fitted models with coefficients, SEs, etc. A set of supplemental tables would be fine.

L208 If you are using models M1-M3 to make this inference, it is highly problematic. You’ve included Site as a random effect in these models which means that any variation among sites in roost size would be mostly captured by the random effect and not by roost size. As a result, you’re basically just analyzing variation in roost size within sites over time and variation in prevalence over time is already captured by your Sampling Period predictor. In short, you didn’t find a roost effect because you included other variables that make it redundant and thus non-significant.

Here's a super simple simulation in R that shows this:

set.seed(5)

df=data.frame(mRS=rep(10^rnorm(17,3,1.),each=15),site=rep(LETTERS[1:17],each=15),

month=rep(1:15,17) ) #17 sites, 15 points each, with mean roost size varying

df$tRS=rnorm(nrow(df),df$mRS,.2*df$mRS) # add temporal variation around mean roost size

ggplot(df,aes(x=month,y=tRS,color=site))+theme_few()+ #plot roost size vs time for sites

geom_point()+geom_line()+

scale_y_continuous(trans="log10")

df$prev=rnorm(nrow(df),.3+.02*log10(df$tRS),.1) #make prevalence a function of roost size

df$N=300 #sample size for bats sampled

ggplot(df,aes(x=tRS,y=prev,color=site))+theme_few()+ #plot of data

geom_point()+

scale_x_continuous(trans="log10")

#Fit model with random effect for site: no significant effect of roost size

f1=lme4::glmer(prev~log10(tRS)+(1|site),family=binomial,weights=N,data=df);summary(f1)

#Fit model without random effect for site; roost size is highly significant

f1=glm(prev~log10(tRS),family=binomial,weights=N,data=df);summary(f1)

L209 Report the P-value. Don’t write <0.01.

L210-12 I think you’ve misinterpreted this analysis. The negative correlation with roost size arises because roost sizes are highest just after birth when juveniles aren’t yet infected. It’s not because of low prevalence of HSV in juveniles – you already have Age in model M3 so that accounts for age differences. It’s because you are examining a dynamic pattern in a static way that doesn’t account for the fact that juveniles become infected as they are also dying/leaving colonies so there is a negative correlation b/w roost size and infection even though there may be no actual biological relationship between the two variables.

The analyses would be much informative if you were modeling the change in infection over time and included roost size (i.e. density dependent transmission) in that analysis. See, for example, ref 11.

Table S1 The font in this table is way too small. Make it bigger.

Table S1 How can a random effect be “significant”? (i.e. Site in M2). What is this based on?

L214 What is “sexual segregation level”? Why would you expect prevalence to vary with sex ratio? Does it vary by sex?

L229 Show this in a figure. You could do this in Fig 2A by showing adult prev by sex.

L237 If the difference is not significant then you should not comment on the difference. If you flip a coin 10 times and get heads 6 times you wouldn’t say the coin trends towards more heads than tails. Also, you need to show the full statistical analysis for this model as for models above.

L250/Fig 3 Can you show the expected patterns of coinfection based on the prevalence of each of the three pathogens? Given the extremely high prevalence of HSV and moderate prevalences of Lepto and PMV we’d expect a large amount of coinfection.

L253-5 The grammar of this sentence needs extensive revision.

L259 Tukey, not Tuckey

L290-1 I don’t understand this sentence and it seems to contradict the one before it.

L303 You’ve actually shown extremely little data on “circulation”. Patterns of infection are not patterns of acquiring infection. Only your multinomial analyses address circulation and so little attention was paid to them that it’s not clear what can be concluded.

L306 “consistent” not “coherent”

L314 No, this is because you didn’t do the analyses properly. See above.

L316-9 This doesn’t make any sense. If high risk individuals roosted in high density areas you WOULD see a correlation with density.

L319-20 Actually you did find evidence of synchrony. Moderately strong for one pathogen, but less so for others. The reason for this is likely the very short time series of your study (2 years), and not much variability in prevalence. In short you’d need much more data to detect synchrony and your vary limited temporal sample cannot be used to suggest synchrony isn’t present.

L321-4 Revise this after you’ve done the analyses properly.

L333 Why “almost”? I thought 100% of HSV+ bats were + on recapture.

L339 You have no data to determine if bats clear the pathogen and become reinfected or simply stop shedding and then start shedding again (recrudescence). Revise this sentence accordingly.

Figure 4C This is the least informative way to show this data and Figure 4 B adds no information and should be removed. In 4C show the 0s and 1s which will provide the same information as the rug plot. In addition, bin the data either into equal sample size bins or roughly equal time intervals and show the prob of that transition (or lack thereof) and SE/CI for each point. This will allow the reader to actually see the data the model is based on.

Figure S2 The %s here don’t add up to 100 so the figure legend should indicate these are the percentages of bats infected with at least one pathogen and the remainder (give the %) were not shedding any of the 3 pathogens. For each section of the venn diagram please add the expected % given the prevalence for each of the three pathogens. My quick calculations are not consistent with the analyses in the paper.

Fig S3 The violin plots on this figure make no sense. They extend to negative time intervals. Please correct. Instead of showing nonsensical violin plots why not show the mean and SE/CI?

L350-4 There are multiple empirical papers showing viral recrudescence that should be cited instead of the papers cited here (the modeling study and hypothesis paper, refs 58, 63). Your Ref 11 (Epstein et al) and these two other papers show strong evidence of recrudescence:

A. R. Sohayati et al.; Henipavirus Ecology Research Group, Evidence for Nipah virus

recrudescence and serological patterns of captive Pteropus vampyrus. Epidemiol.

Infect. 139, 1570–1579 (2011).

A. J. Peel et al., Henipavirus neutralising antibodies in an isolated island population

of African fruit bats. PLoS One 7, e30346 (2012).

L364-9 The absence of a pattern for HSV is likely because prevalence is so high. You don’t need to claim that the effect of reproduction on shedding is “variable” because you didn’t find an effect for HSV.

L380 This sentence contradicts the one before it.

L383 You’ve just shown data that support the opposite of this.

L388 This seems to be more likely to be an age effect than an effect of reproduction. Since shedding prevalence is 5-10x lower for juveniles it’s likely that the non-reproductive individuals are likely just younger and the effect may be unrelated to reproduction. The same may be the case for females.

L429 You didn’t measure infection. You measured pathogen shedding. Revise the wording here and throughout the manuscript.

L431 Antibody levels likely have nothing to do with shedding. HSV often hides out in nerves where antibodies are ineffective. It’s likely that tissue tropism rather than antibody levels that likely explain HSV prevalence.

L436 It’s clearly not “quasi-total” since prevalence wasn’t 0.

L422-442 You have no data on antibodies so I don’t understand why you are speculating so wildly about it. Please remove this entire section of text and focus on the data you have actually collected.

L456-8 This isn’t consistent with Fig S2. This figure appears to show that

L469 You have NOT shown latent herpes virus infection. You’ve shown the opposite. Not a single bat that was shedding upon first capture was not shedding later. This means you have evidence against latent infections for HSV. Revise throughout the paper.

L471-2 Revise after doing the analyses properly.

L473 You have no data on this. Remove.

L518 What pathogens do these primers detect? A single virus for HSV and PMV? Multiple viruses? This is hugely important for interpreting patterns of shedding. Also, is there a reason you didn’t use qPCR? Quantitative estimates of shedding would have been extremely valuable.

L544 These analyses are highly problematic. See comments above.

L575 What data were used to assess segregation? I didn’t see any description of how you measured this.

L578 This is an incorrect way to assess this. Why not include it in your GAMMs?

L584 Spearman tests have very poor power. It’s no wonder you found weak evidence for synchrony. Why not use a more powerful test?

L601 Rstudio is simply a GUI for R. Cite the version of R you used instead.

PLOS authors have the option to publish the peer review history of their article (what does this mean? ). If published, this will include your full peer review and any attached files.

**Do you want your identity to be public for this peer review?** For information about this choice, including consent withdrawal, please see our Privacy Policy .

Reviewer #1: No

Reviewer #2: No
---

## [Decision Letter · Decision Letter 1]

PPATHOGENS-D-24-01674R1

Role of individual and population heterogeneity in shaping dynamics of multi-pathogen shedding in an island endemic bat

PLOS Pathogens

Dear Dr. Aguillon,

Thank you for submitting your manuscript to PLOS Pathogens. After careful consideration, we feel that it has merit but does not fully meet PLOS Pathogens's publication criteria as it currently stands. Therefore, we invite you to submit a revised version of the manuscript that addresses the points raised during the review process.

Please submit your revised manuscript within 30 days Jul 27 2025 11:59PM. If you will need more time than this to complete your revisions, please reply to this message or contact the journal office at plospathogens@plos.org. Please include the following items when submitting your revised manuscript:

We look forward to receiving your revised manuscript.

Kind regards,

Katia Koelle

Guest Editor

Ronald Swanstrom

Section Editor

PLOS Pathogens

Sumita Bhaduri-McIntosh

Editor-in-Chief

PLOS Pathogens

orcid.org/0000-0003-2946-9497

Michael Malim

Editor-in-Chief

PLOS Pathogens

orcid.org/0000-0002-7699-2064

**Additional Editor Comments :**

Thank you for your extensive revision of the manuscript, especially given the detailed comments of Reviewer 2. Reviewer 1 has some remaining minor comments that will help clarify several points in your paper. Please address these and return your manuscript to PLOS Pathogens.

**Journal Requirements:**

1) Regarding Figure 1, thank you for stating “Maps were created using Qgis [111] with a background of OpenStreetMap France under License CC- BY- SA 2.0.”

Please provide a direct link to the base layer of the map (i.e., the country or region border shape) and ensure this is also included in the figure legend.

**Reviewers' Comments:**

Reviewer's Responses to Questions

**Part I - Summary**

Reviewer #1: I am satisfied with the authors' responses to my questions, and cannot comment further on the modelling.

This really is incredible data. I look forward to seeing future work from this system

**Part II – Major Issues: Key Experiments Required for Acceptance**

Reviewer #1: None

**Part III – Minor Issues: Editorial and Data Presentation Modifications**

Reviewer #1: R1.1 – This is an informative and thoughtful response. I appreciate the efforts and it is great to hear the exciting work being done. I appreciate the extra text added, however still feel that it’s a bit obscured. Perhaps the authors could add a note at the end of the first paragraph of the results or to the caption of Table 1 flagging that detections for each pathogen are broad results family/genus level (as appropriate) and may represent combined prevalences across multiple species.

This may not help the authors at this stage, but regarding the substitution of infection for shedding throughout, I agree with reviewer 2 that it is important to mechanistically consider the difference between the two terms, and interpret the results accordingly. However, I think that co-infection is a widely used term to denote an individual from whom two infectious agents are simultaneously detected. Rarely do any infection studies—in wildlife or humans—go to the effort of distinguishing between infection and shedding. While less can be said about infection status in the absence of detection of shedding, the positive detection of the infectious agent more reliably reflects current or very recent infection. With that in mind, I have no problem with the term co-infection. I think co-shedding is an unnecessarily overcomplicated alternative and does not help with clarity. Perhaps the editor can provide a view here to guide the authors.

I may have added a few additional points that I didn't pick up on the first time, but these are minor.

Line 64: "Reproductive individuals (both during the pregnancy and mating)"

| Suggest “Reproductively-active individuals (during the pregnancy, lactation and mating periods)” I am reminded later on that this is clarified in the methods, but this needs to be clear in the abstract too

Line 65:

| Should be associated ‘with’ rather than 'to'. This is a repeated grammatical error in the author summary

Line 65:

| ‘could’ rather than ‘can’

Line 101: co-infections are common in bats [16–20].

| also see meta analysis in Jones et al 2023 Viruses

Line 206: Roost loadings varied only substantially in both magnitude and direction.

| This analysis is not in my expertise, but the placing of the word 'only' seems to make this sentence unclear.

Line 218:

| "see below"

Line 252: During pregnancy

| For clarity, "During the pregnancy period,..."

Line 296-298:

| These two sentences are a little unclear, and I think need restructuring. Either model on the subsequent sentence for HSV, or something like: All captured bats were marked (by tattooing on the wing) allowing to collect shedding data from recaptured bats, and enable comparison of results from samples collected on initial capture and on recapture. XX bats were captured more than once. We tested 471 and 457 samples from XX recaptured bats for PMV and LEPTO, respectively.

Line 304:

| Check the y-axis labels in Fig S3B. Should this be time interval?

Line 322: remained non-reproductively active

| I got to here and was a bit unclear what this means for both females and males. I see now that it is in your methods below, but it would be helpful if you could briefly define "reproductively active" for males and females e.g. ..reproductively active (pregnant, lactating or post-lactating adult females, or adult males with large testes) ...

Line 396-398: Interestingly in D. rotundus infected with a betaherpesvirus [14] , the length of the active period (when shedding occurs) appears to be longer than the latency state (when shedding resumes)

| This sentence is unclear. Latency isn't usually associated with shedding. I would have thought active = shedding and latency = no shedding. Also, note that the study being cited is a modelling study - it should be made clear that there is no empirical evidence to support this point, but that it was a feature of the best fitting mechanistic model

Line 399: although we cannot exclude continuous shedding from persistent infections.

| I think this hypothesis is undersold here. Given that all individuals were shedding on recapture, this should be presented at least as an equal scenario, rather than a caveat

Line 416-418: indicating that cycles of clearance followed by reinfection may occur within the bat population, although the possibility of recrudescence from latent infections cannot be ruled out.

| Similarly, I think these are equally plausible scenarios and should be presented as such

Line 419: bacteria

| because the bacteria?

Line 452: Our results suggest that pregnancy increases the probability of PMV and LEPTO shedding, as well as the likelihood of dual and triple shedding in Reunion free - tailed bats .

| The results suggest that this was clouded by age though, considering M0 females? The nuance of the M0 results was striking to me in the results section, but I think the discussion of those results here is not faithful to them. To me, the results showed clearly that these effects were not reproduction effects but were age effects. Here, the discussion is focussed on the reproduction results, with the age as a small caveat. I really think that the age results should be considered alongside the reproduction results. i.e. while reproduction is often found to be a significant predictor, and many of our results here supported this, our ability to separate non-juvenile individuals into those that had previously bred (reproductive adults) and those who had not (likely younger pre-reproductive individuals), the effects of reproduction disappeared.

Line 494: the timing

| This is an important point to get across clearly. It took be a couple of reads to follow. Perhaps emphasis this point as 'asynchronous timing'

PLOS authors have the option to publish the peer review history of their article (what does this mean? ). If published, this will include your full peer review and any attached files.

**Do you want your identity to be public for this peer review?** For information about this choice, including consent withdrawal, please see our Privacy Policy .

Reviewer #1: No

**Figure resubmission:**
---

## [Editor Report · Decision Letter 2]

Dear Dr Aguillon,

We are pleased to inform you that your manuscript 'Role of individual and population heterogeneity in shaping dynamics of multi-pathogen shedding in an island endemic bat' has been provisionally accepted for publication in PLOS Pathogens.

Best regards,

Ronald Swanstrom

Section Editor

PLOS Pathogens

Ronald Swanstrom

Section Editor

PLOS Pathogens

Sumita Bhaduri-McIntosh

Editor-in-Chief

PLOS Pathogens

orcid.org/0000-0003-2946-9497

Michael Malim

Editor-in-Chief

PLOS Pathogens

orcid.org/0000-0002-7699-2064
---

## [Editor Report · Acceptance letter]

Dear Dr Aguillon,

We are delighted to inform you that your manuscript, "Role of individual and population heterogeneity in shaping dynamics of multi-pathogen shedding in an island endemic bat," has been formally accepted for publication in PLOS Pathogens.

Best regards,

Sumita Bhaduri-McIntosh

Editor-in-Chief

PLOS Pathogens

orcid.org/0000-0003-2946-9497

Michael Malim

Editor-in-Chief

PLOS Pathogens

orcid.org/0000-0002-7699-2064